# Detection and recognition of metal surface corrosion based on CBG-YOLOv5s

**Mingjiao Fu[1], Zhitong Jia[1], Lingzhi Wu[2], Zhendong Cui** [1,2]*

1 School of Computer and Control Engineering, Yantai University, Yantai, China, 2 Dongfang Electronics Corporation, Yantai, China

* cuizhendong@126.com

## Abstract

The automatic detection of the degree of surface corrosion on metal structures is of significant importance for assessing structural damage and safety. To effectively identify the corrosion status on the surface of coastal metal facilities, this study proposed a CBG-YOLOv5s model for metal surface corrosion detection, based on the YOLOv5s model. Firstly, we integrated the Convolutional Block Attention Module (CBAM) into the C3 module and developed the C3CBAM module. This module effectively enhanced the channel and spatial attention capabilities of the feature map, thereby improving the feature representation. Second, we introduced a multi-scale feature fusion concept in the feature fusion part of the model and added a small target detection layer to improve small target detection. Finally, we designed a lighter C3Ghost module, which reduced the number of parameters and the computational load of the model, thereby improving the running speed of the model. In addition, to verify the effectiveness of our method, we constructed a dataset containing 6000 typical images of metal surface corrosion and conducted extensive experiments on this dataset. The results showed that compared to the YOLOv5s model and several other commonly used object detection models, our method achieved superior performance in terms of detection accuracy and speed.

**Data Availability Statement:** The data set underlying this article has been uploaded to Figshare and is accessible via. https://figshare.com/s/63accf67b33154b291ed.

**Funding:** The author(s) received no specific funding for this work.

## 1. Introduction

With the rapid advancement of science and technology, the marine economy has increasingly become a significant component of China's national economy [1,2]. A large number of nearshore and offshore facilities have been put into operation, with steel being the primary material required for these facilities. However, the marine environment is one of the most corrosive natural environments. The marine atmosphere contains various salt particles that have a strong corrosive effect on metals. Fig 1 illustrates the corrosion status of metal facilities in a marine environment. Metal corrosion leads to material loss and wastage, thereby causing substantial economic losses. According to a 2015 research report by the Chinese Academy of Engineering, economic losses caused by corrosion in China amount to approximately USD 310 billion, equivalent to 3.34% of GDP [3]. Furthermore, metal corrosion reduces the mechanical properties of metal materials such as strength, toughness, hardness etc., rendering

**Competing interests:** The authors have declared that no competing interests exist.

**Fig 1. Corrosion of metallic facilities in the marine environment.**

them incapable of withstanding normal workloads and potentially leading to failures such as fractures or deformations. For critical equipment and structures like bridges, pipelines, aircrafts and trains this poses severe safety threats and could even result in catastrophic consequences. Simultaneously, metal corrosion generates substantial waste and toxic substances (like rust, verdigris and lead salts). These substances could infiltrate into the environment via water flow or air flow or soil seepage polluting water sources soil and air thereby posing threats to human health as well as flora and fauna. Therefore conducting rapid accurate corrosion identification grading detection on steel facilities in marine atmospheric environments holds positive significance for relevant personnel to adopt appropriate anti-corrosion measures thereby avoiding further serious corrosion damage to equipment.

The traditional evaluation of the degree of corrosion on metal structures primarily relies on manual visual methods. This approach not only consumes a significant amount of manpower and time, but long periods of inspection can also lead to operator fatigue. Furthermore, due to a lack of consistency in the evaluation criteria and susceptibility to subjective influences, the accuracy of the determination is limited. As a result, researchers have successively proposed various methods for detecting metal corrosion. Currently, methods for detecting corrosion on metal surfaces can be classified into two main categories: one is based on physical or chemical principles, and the other uses computer vision technology for detection.

Detection methods based on physical or chemical processes primarily use the physical or chemical changes that occur on the metal surface during the corrosion process to assess the condition and degree of metal corrosion. These methods include electrochemical methods, resistance methods, ultrasonic waves, and spectroscopy, among others. Compared with traditional manual visual methods, these detection methods can directly measure the corrosion parameters of the metal surface, such as potential, current, resistance, sound speed, and magnetic field and spectrum, and reflect the corrosion status of the metal surface from these parameters. However, these detection methods require professional instruments and operator.

With the constant advancement of computer technology, computer vision has been extensively applied to the field of corrosion detection. Researchers have begun to utilise the computer vision technology for classification and object detection of corrosion images. This allows for an intuitive presentation of the corrosion morphology and distribution on the metal surface, and enables efficient and accurate processing of a large amount of image data, thereby enhancing detection efficiency and accuracy. Although traditional detection methods based

on computer vision have seen some improvements, their accuracy has not yet reached the desired level, and their operation is not sufficiently convenient. Deep learning has brought new ideas to object detection. Convolutional neural networks [4] can extract high-level features, thereby significantly improving classification accuracy. However, current object detection approaches based on deep learning suffer from issues such as loss of metal corrosion information due to model training on open source datasets, large model size, and inability to meet practical application requirements.

Therefore, we selected the most widely used YOLOv5s model in the YOLO series [5–8] as the basis and designed a more lightweight CBG-YOLOv5s object detection model. This model has the benefits of small size, high speed and high accuracy, which theoretically supports deployment on embedded platforms.

The study's main contributions can be summarised as follows:

1. We have developed a corrosion detection model for metal surfaces, called CBG-YOLOv5s. This model can classify metal surfaces into three corrosion levels based on their texture, colour, and depth of the corrosion. It can assist technicians in accurately and promptly identifying the corrosion level of metals.

2. We collected 600 original images of corroded metal surfaces and performed data augmentation, constructing a dataset of metal surface corrosion images containing 6000 images.

3. In order to improve the detection accuracy of the model, we introduced the C3CBAM module and C3Ghost module, expanded the scale of the YOLOv5s model, and added a small target detection layer. Compared with several other commonly used object detection models, our method had achieved superior performance in terms of detection accuracy and speed.

## 2. Related work

In the early stages of the research, corrosion detection primarily relied on physical or chemical methods. Yeih et al. [9] used the amplitude attenuation method of ultrasonic detection technology to assess damage from metal corrosion in reinforced concrete, achieving significant detection results. Hong et al. [10] proposed a method for detecting the detachment of the outer protective layer of underwater metal pipes using ultrasound imaging technology based on Support Vector Machine (SVM) and Histogram of Oriented Gradients (HOG) techniques. However, this method did not perform well in detecting narrow detachment damage areas. Wicker et al. [11] proposed a relatively economical method for non-destructive detection of metal corrosion based on simple data analysis and infrared thermal imaging technology. To effectively assess the degree of wire corrosion, Li et al. [12] used Acoustic Emission (AE) technology to detect the externally applied current cathodic protection (ICCP) and prestress changes in the marine atmospheric rainwater environment, and based on this, assessed the degree of rebar corrosion.

In the field of metal corrosion detection, image processing technology has achieved significant detection results. For instance, Pakrashi et al. [13] proposed a method for detecting the degree of metal surface corrosion based on regional optical contrast. This method analyzed the optical contrast between the corrosion area and the surrounding environment and combined it with image processing technology for edge detection. However, it was mainly applicable to pitting corrosion on aluminum surfaces. Additionally, Shen et al. [14] proposed a corrosion recognition method that combined color image processing technology with Fourier transform, based on texture features and color features. This method showed a clear advantage in dealing with non-uniform illumination conditions. Ghanta et al. [15] evaluated corrosion defects on

the surface of steel-coated bridges by using a single-scale Haar wavelet transform on RGB sub-images, but this detection method had high requirements for image quality.

In the field of metal corrosion detection, two main categories of deep learning-based object recognition algorithms have been used: single-stage and two-stage. Two-stage algorithms divide feature extraction and object detection into two steps, mainly represented by Region-based Convolutional Neural Network (R-CNN) [16], Faster R-CNN [17] and Fast R-CNN [18]. For example, Guo et al [19] proposed a Faster R-CNN model with a feature enhancement mechanism. This model used the ResNet-101 residual network as the backbone network and implemented a feature enhancement mechanism after the Region of Interest (ROI) pooling layer to achieve the goal of detecting rust on transmission line fittings, achieving a detection accuracy of more than 97%. Additionally, Tian et al [20] proposed a metal corrosion recognition algorithm based on Faster R-CNN and rust HIS colour features. This algorithm converted the corrosion image into a HIS colour model to determine and annotate the corrosion pixels. The Faster R-CNN model was then used to locate and detect the annotated corrosion areas, achieving high accuracy and recall rates. However, due to the complex network structure of the two-stage object detection models, poor real time performance and the size of the models with high hardware demands, these models are not suitable for subsequent deployment on embedded development platforms.

Optimized single-stage detection methods integrate the two steps of feature extraction and object detection, effectively reducing redundant computations and significantly improving the speed of detection. SSD [21] and the YOLO series are the most representative methods in this field. For example, Ramalingam et al. [22] proposed an enhanced SSD MobileNet framework, which includes a periodic pattern detection filter based on self-filtering, used to detect surface defects on aircraft caused by corrosion and cracks. Mukhiddinov et al. [23] proposed a multi-class fruit and vegetable classification system based on an improved YOLOv4 model, which divided the recognised fruit types into fresh or rotten categories. Deyin et al. [24] proposed an aircraft surface defect detection model based on the YOLO network, used to automatically identify corrosion and various defects on the aircraft surface. Matthaiou et al. [25] achieved good results by training SSD using transfer learning technology to detect corrosion objects. Jia et al. [26] proposed the Corrosion-YOLOv5s metal corrosion target detection model based on the YOLOv5s model, which achieved an accuracy rate of 90.5%.

We have made a thorough analysis and study of previous work, and the advantages and disadvantages of each method are summarised in Table 1. We have learnt from the previous research experience and designed the CBG-YOLOv5s model for this work. Compared with the above research work, our proposed model has improved in terms of detection efficiency and accuracy.

The remainder of this paper is organised as follows: Section 3 gives a detailed description of the basic model used in this study. Section 4 discusses the proposed model in more detail and the corresponding improvement measures. Section 5 describes in detail the process of preparing the dataset. The experimental results of the model are mainly shown in section 6 and the experimental results are discussed. Finally, Section 7 summarises the entire paper and looks forward to future work.

## 3. YOLOv5 model

YOLOv5 is a widely used single-stage object recognition algorithm. Based on the depth and width of the model, the YOLOv5 series is classified into YOLOv5x, YOLOv5l, YOLOv5m and YOLOv5s. Considering that the use of a model with a huge number of parameters on embedded devices could have adverse effects, this study chose the lighter YOLOv5s as the benchmark

**Table 1. Advantages and disadvantages of different corrosion detection methods.**

| Category | Advantages | Disadvantages | Related work |
|---|---|---|---|
| **Physical detection methods** | High detection accuracy without affecting the performance and service life of the material. | High cost of instrumentation and high operating skill requirements. | Yeih et al. [9] Hong et al. [10] |
| **Traditional image processing techniques** | Fast computation, simple algorithms, no need for large amounts of training data. | Sensitive to image quality and lighting conditions, difficult to handle complex backgrounds and occlusions, low accuracy and robustness. | Pakrashi et al. [13] Shen et al. [14] Ghanta et al. [15] |
| **Two-stage target detection methods** | High detection accuracy. | Complex network structure, poor real-time, and the large size of the model has high hardware requirements. | Guo et al. [19] Tian et al. [20] |
| **One-stage target detection methods** | Effectively reduces redundant computations, giving the algorithm a significant advantage in detection speed. | In the localisation and detection of small targets, due to the high number of densely generated candidate frames, it is easy to produce a high false detection rate. | Ramalingam et al. [22] Deyin et al. [24] Matthaiou et al. [25] |

model for metal corrosion detection from a practical point of view. As can be seen in Fig 2, the structure of this model mainly comprises four parts: the input end, the feature acquisition network (backbone), the feature fusion network (neck), and the recognition head (head).

In the YOLOv5 algorithm, the input end adopted Mosaic data enhancement, adaptive image resizing and adaptive anchor box calculation strategies for data pre-processing. The Mosaic data augmentation method randomly cropped a selected image and three other random images and then stitched them together to form a new image. This enriched the background of the image, increased the number of small targets, increased the diversity of the dataset and thereby improved the robust nature of the model. The backbone structure was composed of multiple Conv, C3 and SPPF modules. By mixing multiple Conv and C3 modules, the feature extraction capability of the model was improved. The SPPF module used multiple small-size pooling cores in a cascade to replace the single large-size pooling core in the previous SPPF module, greatly improving the model's feature extraction capability. This improvement helped to detect target objects of different sizes and further accelerated the model's operation. In the neck part, the YOLOv5 model adopted the PANet structure and added a bottom-up path enhancement structure to the top-down feature pyramid to improve the

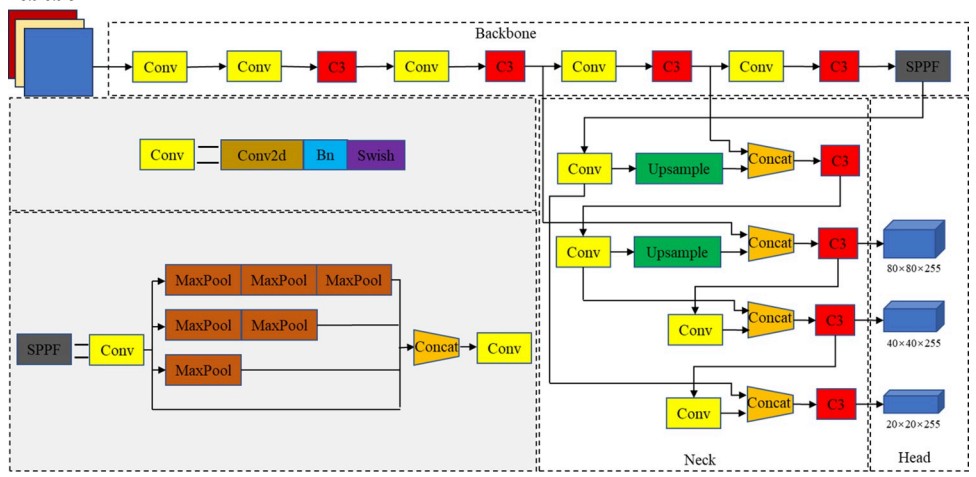

**Fig 2. YOLOv5s model structure framework.**

network's feature fusion capability. The head part contained three detection layers corresponding to three distinct sizes of feature maps acquired from the neck part. According to the size of the feature map, a grid was divided on the feature map, and for each grid, three anchors with different aspect ratios were preset for target prediction and regression.

YOLOv5 employed CIoU_Loss [27] as the loss function for the bounding box regression, as shown in Eq 1. In this context, Eq 2 defined the weighting coefficient, Eq 3 was used to gauge the proportionality between two rectangular boxes, and Eq 4 was used to obtain the ratio of the union of the intersection of the predicted box and the real box.

$$L_{CIoU} = 1 - IoU + \frac{\rho^2(b, b^{gt})}{c^2} + \alpha v \tag{1}$$

$$\alpha = \frac{v}{(1 - IoU) + v} \tag{2}$$

$$v = \frac{4}{\pi^2} \left( arctan \frac{w^{gt}}{h^{gt}} - arctan \frac{w}{h} \right)^2 \tag{3}$$

$$IoU = \frac{A \cap B}{A \cup B} \tag{4}$$

In this context, $b$ and $b^{gt}$ denote the centroids of the actual and predicted boxes respectively, $\rho$ denotes the Euclidean separation of these two rectangles, $c$ denotes the diagonal separation of the area contained by these two rectangles, and $w$, $h$ and $w^{gt}$, $h^g$ denote the width and height of the predetermined and actual boxes respectively.

## 4. CBG-YOLOv5s model

In this study, we designed the CBG-YOLOv5s object detection network model based on the YOLOv5s version, as shown in Fig 3. In response to issues such as high background complexity of the corrosion area on the metal surface in the dataset, small category differences, and large dataset density and parameter volume, we made improvements in three aspects.

### 4.1. Convolutional attention mechanism module based C3CBAM model

In the field of vision, the attention mechanism played a key role. It gave neural networks the ability to generate masks, allowing them to automatically learn and focus on important areas. By performing weighted iterations based on mask scores, it was possible to increase the influence of the focus area and reduce the weight of irrelevant information, thereby optimising the network model's performance. Depending on the location of the mask generation, the attention mechanism could be broadly classified into three types: channel attention mechanism, spatial attention mechanism, and mixed-domain attention mechanism. In this study, a mixed-domain attention mechanism, the CBAM module [28] was adopted. The CBAM module included a channel attention module (CAM) and a spatial attention module (SAM), the detailed structure of which is shown in Fig 4.

The working process of the CBAM module was as follows: Initially, the input feature $F \in R^{C*H*W}$ was processed by the CAM module, generating $M_c \in R^{C*1*1}$, which was the channel weight vector. Then, $M_c$ was multiplied with $F$, resulting in the weighted feature $F'$. Subsequently, $F'$ was input into the SAM module, yielding the spatial weight matrix $M_s \in R^{1*H*W}$.

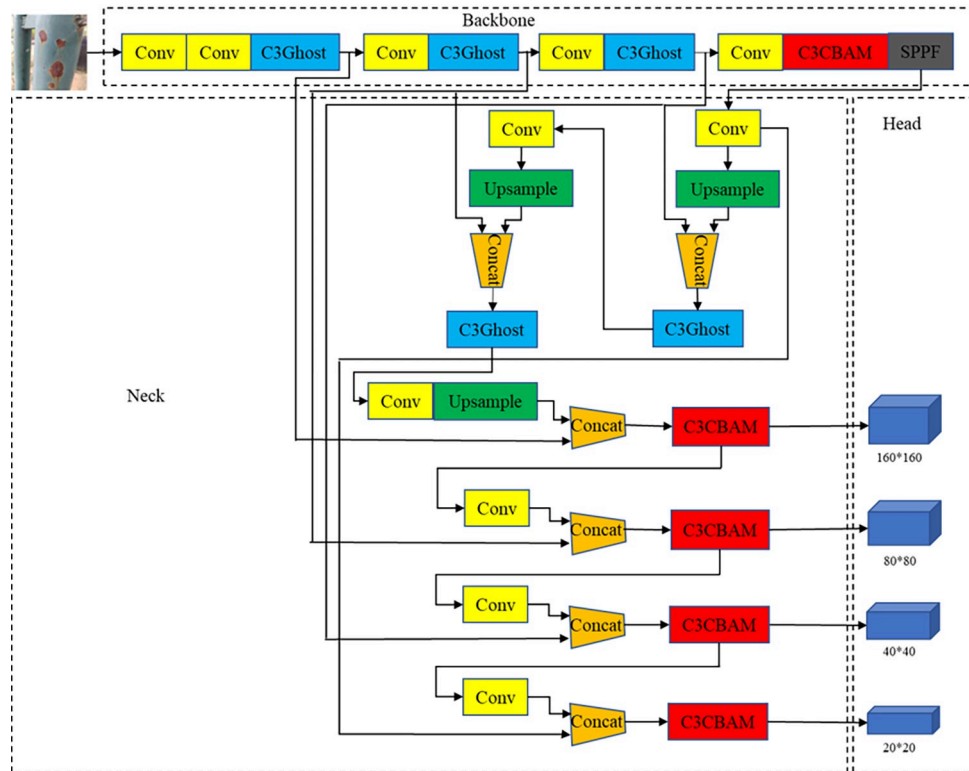

**Fig 3. CBG-YOLOv5s model structure framework.**

Finally, $M_s$ was multiplied with $F'$, producing the spatially weighted feature $F''$.

$$F' = M_c(F) \otimes F \qquad (5)$$

$$F'' = M_s(F') \otimes F' \qquad (6)$$

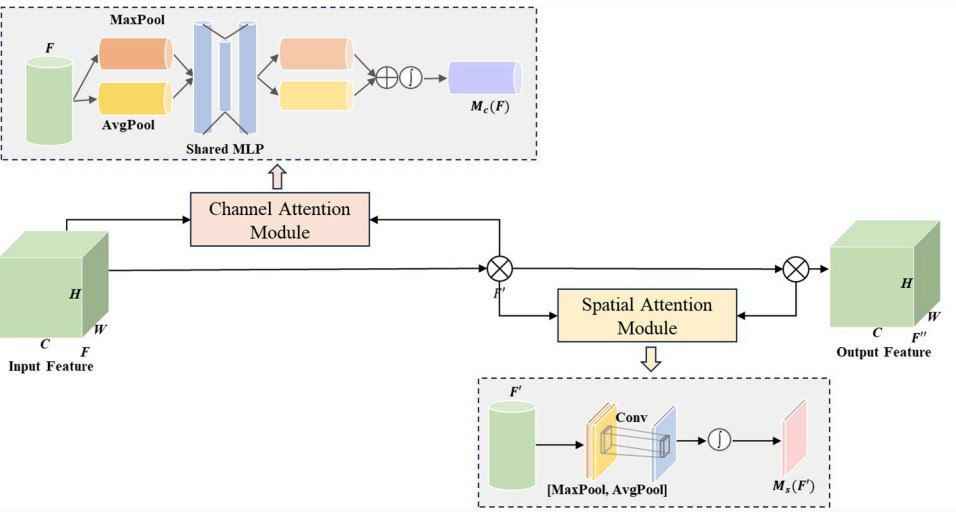

**Fig 4. CBAM model structure framework.**

The CAM module worked by focusing on useful feature channels and ignoring useless ones, allowing the model to concentrate more on effective spatial information. Initially, we employed two parallel operations, namely average pooling and max pooling, to integrate the spatial information of the feature map *F*. Subsequently, these two types of features were passed into a Multilayer Perceptron (MLP), and through parallel max pooling and average pooling operations as well as non-linear activation function processing, the output result of the CAM was obtained. The calculation formula for CAM was as follows:

$$M_c(F) = \sigma(MLP(AvgPool(F)) + MLP(MaxPool(F))) \tag{7}$$

By introducing the SAM module, the model was able to focus more on the areas of interest in the feature map. Initially, we used two parallel operations, average pooling and max pooling, to integrate the channel information in the feature map *F'*. Then, these two types of features were connected and convolved through a convolutional layer, resulting in the final output of the SAM. The calculation formula for SAM was as follows:

$$M_s(F') = \sigma(f([AvgPool(F'); MaxPool(F')])) \tag{8}$$

Here, $\sigma$ represents the sigmoid function, and *f* represents a convolution operation with a filter size of 7×7.

The CBAM attention mechanism enabled the network to focus more on significant features and suppress unimportant ones, thereby guiding the network to pay attention to which features and their locations, further improving the accuracy of corrosion area positioning and detection. We have combined the C3 module with the CBAM module to form the C3CBAM module. By adding this module to the SPPF layer before the backbone part and the neck part, the expressiveness of the features and the multi-scale fusion ability were further enhanced, thus significantly increasing the accuracy and speed of detection.

## 4.2. BiFPN-CBAM weighted bi-directional feature fusion network based on fusion attention mechanism

In the target detection algorithm field, to solve the problem of image feature loss due to the increase of network layers, we usually adopt the method of constructing a feature pyramid to fuse semantic information at different levels. In the YOLOv5 network, we adopted the Path Aggregation Network (PANet) structure [29], which simply fused the features of the third to fifth layers. In order to fully account for the impact of different resolutions of the incoming feature map on the outgoing feature map, the Google team proposed a bidirectional weighted BiFPN structure [30] and improved on the basis of PANet. BiFPN fused the features of layers 3 to 7, and deleted the nodes of layers 3 and 7 with small contributions to reduce computation. At the same time, it introduced a cross-scale connection method to avoid too much loss of deep semantic information while preserving shallow linguistic information.

This study drew inspiration from the design philosophy of BiFPN, adding two cross-scale connections between input and output nodes of the same scale to optimize the detection effect of target-dense images. Simultaneously, to enhance the model's detection capability for small target data, we incorporated the second feature layer into the feature fusion network to retain shallow information, and increased the number of output detection layers to four and expanded the number of prediction boxes from nine to twelve. This method broadened the model's perceptual range, heightened its sensitivity to small targets, and further improved the detection effect of small targets.

The complexity and diversity of the background of metal corrosion, such as the inconsistency in the shooting distance of manually collected images, and the differences in corrosion color and texture caused by different metals in different environments, made it difficult for the

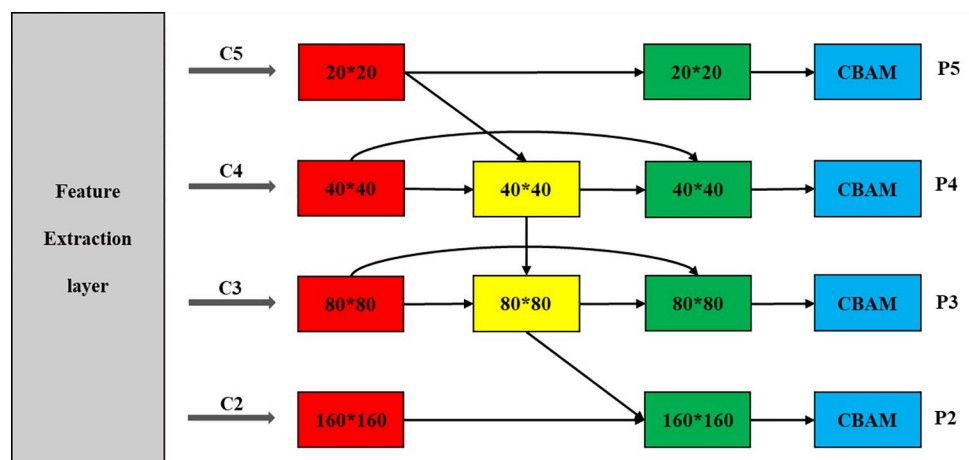

**Fig 5. BiFPN-CBAM structure framework.**

model to extract high-quality information, thereby increasing the missed detection rate and false detection rate of the corrosion area. To effectively solve these problems, we adopted a strategy to enhance the perceptual ability of the model and further optimized it. Based on the feature fusion network we proposed, we introduced the CBAM module. In this way, YOLOv5's feature fusion network can not only enhance the model's detection ability for small target data, but also strengthen the importance screening of different channel features and the attention to direction and position information. We named this module BiFPN-CBAM, as shown in Fig 5.

Compared to the original YOLOv5 model's PANet feature fusion network, the BiFPN-CBAM we designed demonstrated superior performance when dealing with datasets where factors such as dense targets, complex backgrounds, and low image resolution were present.

## 4.3. C3Ghost model based on lightweight neural network

Ghost Convolution [31], as a lightweight convolutional neural network, successfully reduced the number of parameters and computational complexity while maintaining model performance. Ghost Convolution adopted a two-step strategy to replace traditional convolution operations: firstly, it generated a smaller number of feature maps through ordinary convolution operations; then, it produced more feature maps through linear transformation and combination operations. Finally, it concatenated these two groups of feature maps into a Ghost feature map. Compared to ordinary convolution operations, Ghost Convolution greatly reduced computational load and the number of parameters. As shown in Fig 6, it illustrated the traditional convolution and Ghost Convolution modules.

In order to decrease the model size, in this paper we used Ghost Convolution to replace the ordinary convolution operation in the C3 module, thus constructing a lighter C3Ghost module, the concrete structure of which is shown in Fig 7. In GhostConv, we used a 1×1 convolution kernel to decrease the number of channels of the input feature map to half of the original, and concatenated it with the feature map obtained after processing with a 5×5 convolution kernel. The transformed GhostBottleneck first reduced the number of input feature map channels by half through the first GhostConv, then restored it to the original number of channels through the second GhostConv, and fused it with the features obtained through 3×3 depth convolution. By substituting the bottleneck in the C3 module with GhostBottleneck, the C3Ghost module achieved a reduction in parameters and computational complexity, improved model efficiency and speed, and enhanced the expressiveness of feature maps, thereby improving model accuracy and robustness.

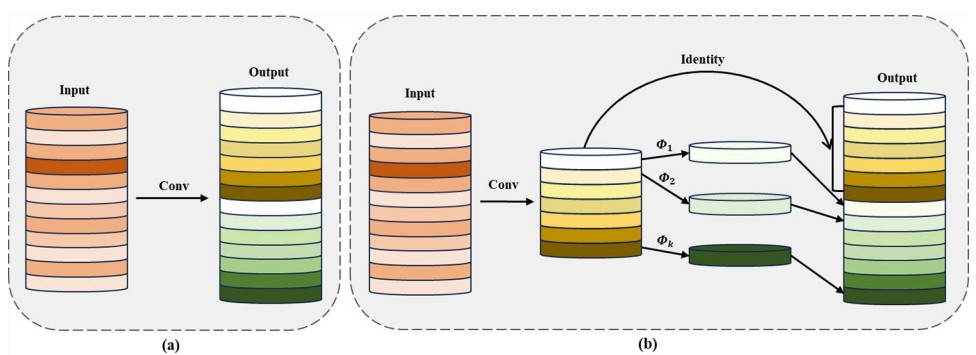

**Fig 6. Convolution contrast diagram.** (a) Ordinary convolution operation. (b) Ghost convolution operation.

## 5. Data set production

The learning results of the model were influenced by the dataset, and the generalisation ability of the model had a certain correlation with the diversity of corrosion images in the dataset. This research aimed to explore the corrosion of metal surfaces under marine environments. The data processing flow of metal surface corrosion is shown in Fig 8.

We selected the coastal area of Yantai as the data collection site due to its typical marine climate characteristics, where seawater had a strong corrosive effect on metal surfaces. To ensure the quality and consistency of the images, we used professional image collection equipment and strictly followed the standards and norms during operation. During the collection process, we obtained a total of 732 images of metal surface corrosion with different types, degrees,

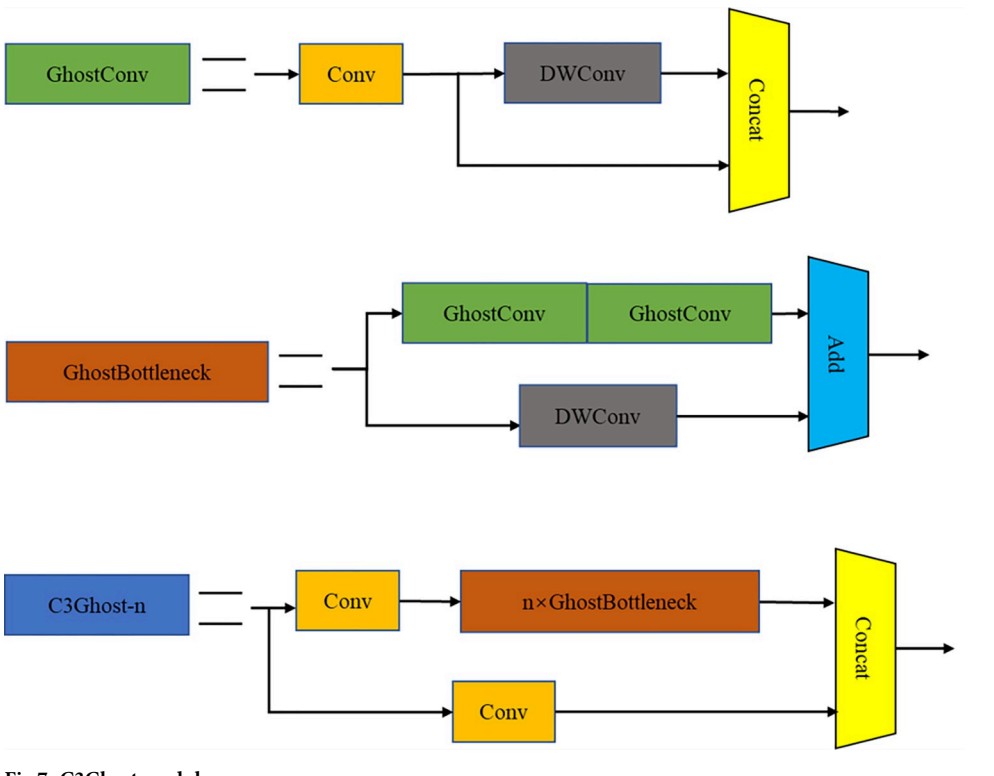

**Fig 7. C3Ghost module.**

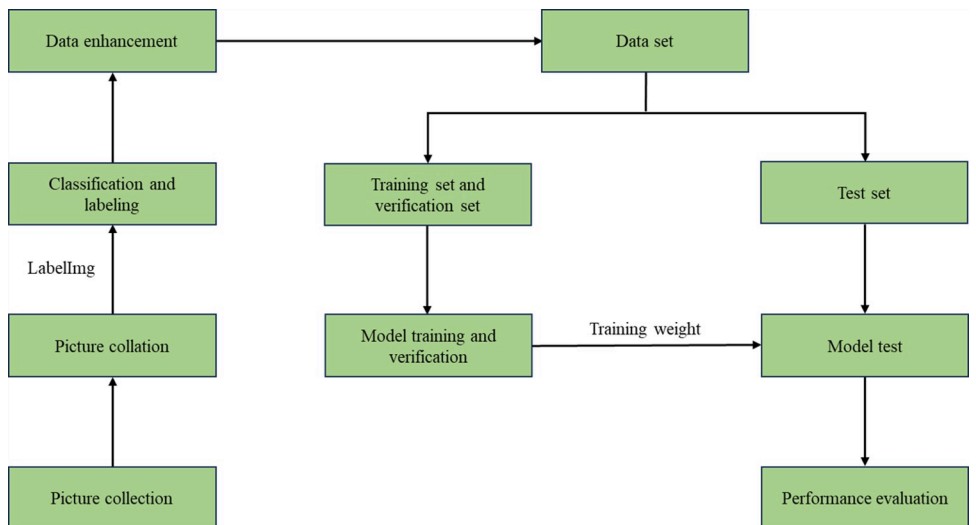

**Fig 8. Processing flow of metal surface corrosion data.**

states, and colors. After removing blurry and duplicate images, we finally selected 600 representative images of metal surface corrosion.

In order to enhance the efficiency of model training and to reduce the computational burden, we standardised the size of the images by adjusting all images to a dimension of 640×640. Based on the colour, texture and other characteristics of metal surface corrosion, we classified it into three types: light corrosion (LC), moderate corrosion (MC) and heavy corrosion (HC). Fig 9 shows the metal surface corrosion conditions of these three types. To ensure the balance of various targets in the dataset, we collected 200 images of metal surface corrosion in each grade and used LabelImg annotation software to accurately annotate the corroded areas on the metal surface.

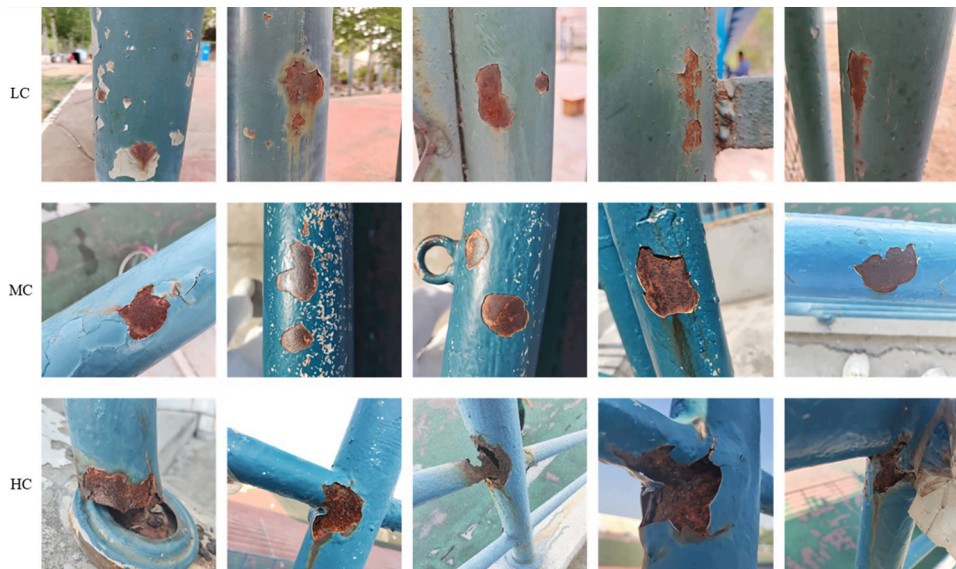

**Fig 9. Corrosion of different types of metal surfaces.**

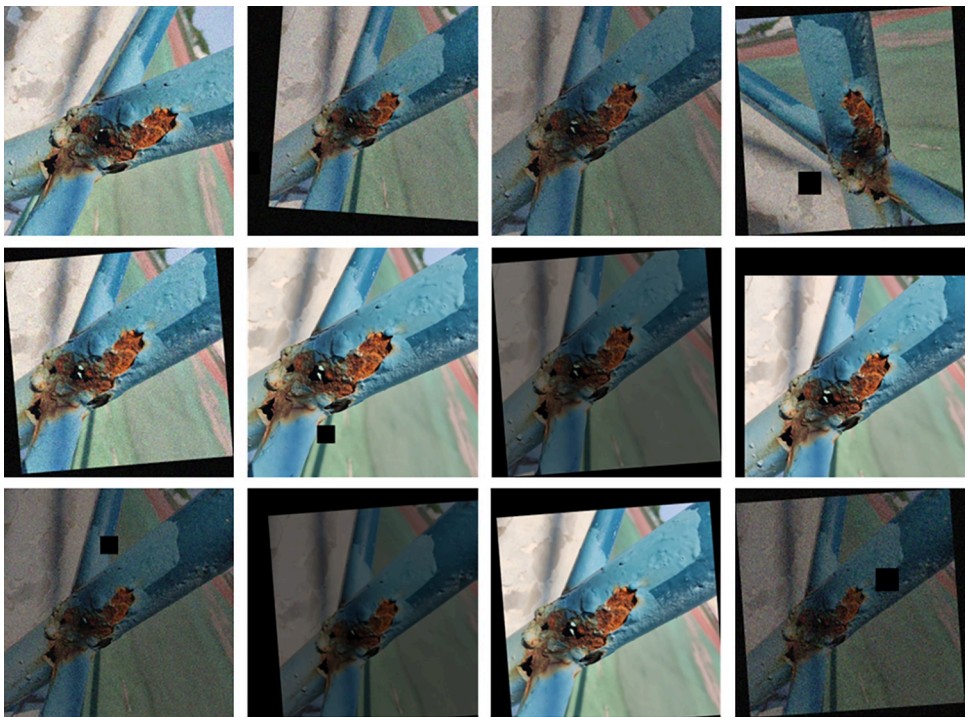

**Fig 10. Demonstration of the effects of data enhancement.**

To increase the robustness of the model, we employed five data augmentation techniques, including cropping, translation, brightness adjustment, noise addition and angle rotation. These techniques were randomly combined to enhance the original images. The enhancement effect is shown in Fig 10. The enhanced images of metal surface corrosion were randomly divided into training, validation and test sets in a ratio of 8:1:1, resulting in a data set of 4800 training images, 600 validation images and 600 test images, making a total of 6000 images. These images were used to construct a metal surface corrosion detection dataset for this study. The specific partitioning of the dataset is shown in Table 2.

## 6. Experiments

### 6.1. Experimental environment and parameter settings

The experimental environment was as shown in Table 3 below.

We set the starting learning rate for the YOLO model to 0.01, the momentum parameter to 0.937, the batch size to 16, and the number of rounds of training to 200.

**Table 2. Division of metal surface corrosion datasets.**

| Class | Train | Valid | Test |
|---|---|---|---|
| LC | 1600 | 200 | 200 |
| MC | 1600 | 200 | 200 |
| HC | 1600 | 200 | 200 |
| all | 4800 | 600 | 600 |

**Table 3. Experiment environment.**

| Parameters | Experiment Environment |
|---|---|
| Operating System | Ubuntu 20.04.4 |
| CPU | Intel®Core™ i5-12400F |
| GPU | GeForce RTX 3060(12GB) |
| Python | 3.8 |
| Deep Learning Framework | pytorch1.11.0, CUDA11.6 |

## 6.2. Evaluation indicators

The aim of this study was to enhance the model's detection accuracy while reducing the model's parameter volume. To test the effectiveness of the proposed model, we quantitatively evaluated the performance of the algorithm using a set of evaluation metrics widely accepted in the object detection field, including accuracy, recall, F1 score, FPS, mAP and parameter volume, which were defined separately using Eqs (9)–(13). Among them, accuracy was used to measure the probability of correctly predicting positive samples; recall represented the proportion of actual positive samples that were correctly predicted; AP was the area under curve based on recall and precision of each category; mAP was the mean value of AP values of each category, the higher its value, the greater the detection accuracy of the model and the more effective; FPS represented the number of frames per second processed by the model, which was used to assess the speed of detection of the model.

$$Precision = \frac{TP}{TP + FP} \tag{9}$$

$$Recall = \frac{TP}{TP + FN} \tag{10}$$

$$F_1 - score = \frac{2 \times precision \times Pecall}{Precision + Recall} \tag{11}$$

$$AP = \int_0^1 P_{(r)} dr \tag{12}$$

$$mAP = \frac{\sum_{i=1}^{c} AP_i}{c} \tag{13}$$

where *TP*, *FP* and *FN* are the number of true positives, the number of false positives and the number of false negatives, respectively, and *c* is the number of classes.

## 6.3. Comparison with YOLOv5s model

Through an in-depth analysis of the comparative experimental results, we found that under the same environmental conditions, the same initial parameter settings, and the same dataset, the CBG-YOLOv5s model achieved significant improvements in all evaluation indicators compared to the original YOLOv5s model (see Tables 4 and 5 for details). Notably, in light corrosion images where there were a large number of small targets, the CBG-YOLOv5s model demonstrated significant performance advantages, which fully validated the effectiveness of our strategy to add a small target detection layer. Further, after a comprehensive comparison of these two models in Table 6, we found that the CBG-YOLOv5s model not only achieved

**Table 4. YOLOv5s model performance.**

| Class | Images | Labels | P/% | R/% | mAP@.5/% | mAP@.5:.95/% |
|-------|--------|--------|-----|-----|----------|--------------|
| LC | 200 | 594 | 81.8 | 84.0 | 89.2 | 68.7 |
| MC | 200 | 432 | 96.3 | 83.7 | 93.2 | 79.3 |
| HC | 200 | 214 | 92.1 | 87.4 | 94.5 | 78.8 |
| All | 600 | 1240 | 90.1 | 85.0 | 92.3 | 75.6 |

significant improvements in key indicators such as accuracy, recall rate, F1-score, mAP@.5 and mAP@.5:.95 (increasing by 3.2%, 1.6%, 2.4%, 2.7% and 1.2% respectively), but also successfully reduced about 1.37 million parameters. These experimental results fully prove that the CBG-YOLOv5s model can not only enhance the detection accuracy, but also reduce the parameter volume by about 20%. Therefore, the proposed improvement strategy has broad prospects of application in the field of metal surface corrosion degree detection.

Upon conducting an in-depth testing of the YOLOv5s model and the CBG-YOLOv5s model, we found that the CBG-YOLOv5s model significantly improved the issues of small target misdetection and omission in complex backgrounds compared to the original model, and demonstrated more accurate positioning capabilities and higher detection accuracy. Notably, the CBG-YOLOv5s model was not only able to accurately detect corrosion areas, but also accurately identify the grades of corrosion areas. The specific test results can be seen in Figs 11 and 12.

## 6.4. Comparison with other target detection models

After an in-depth comparison of the enhanced algorithm with the current leading object detection algorithms (including CenterNet, Faster R-CNN, SDD, YOLOv3, YOLOv4 and YOLOv5s) under the same experimental conditions and using the same dataset, Table 7 demonstrates the comparison results. We found that the CBG-YOLOv5s model has a faster detection speed compared to the CenterNet and Faster R-CNN two-stage object detection models. This is mainly due to the lower complexity of the one-stage object detection algorithm compared to the two-stage object detection algorithm. In addition, the CBG-YOLOv5s model achieved significant improvements in accuracy, recall rate, mAP value and F1 score compared to the SDD, YOLOv3, YOLOv4 and YOLOv5s single-stage object detection models, demonstrating higher detection accuracy. Although the improved CBG-YOLOv5s model had an increased number of network layers due to the addition of a small target detection layer, which reduced its recognition speed, its recognition speed still far exceeded other comparison models, except for being lower than the original version of YOLOv5s. These comparative experimental results fully validate the superior performance of the CBG-YOLOv5s object detection model in metal corrosion target detection.

Fig 13 demonstrated the performance of the CBG-YOLOv5s model in comparison with other models in terms of accuracy in recognizing surface corrosion on metals. It was clearly

**Table 5. CBG-YOLOv5s model performance.**

| Class | Images | Labels | P/% | R/% | mAP@.5/% | mAP@.5:.95/% |
|-------|--------|--------|-----|-----|----------|--------------|
| LC | 200 | 594 | 88.1 | 80.3 | 91.6 | 69.0 |
| MC | 200 | 432 | 96.9 | 87.6 | 95.0 | 79.4 |
| HC | 200 | 214 | 94.7 | 92 | 98.4 | 81.9 |
| All | 600 | 1240 | 93.3 | 86.6 | 95.0 | 76.8 |

Table 6. Performance comparison between YOLOv5s and CBG-YOLOv5s.

| Method | P/% | R/% | F1-score | mAP@.5/% | mAP@.5:.95/% | Params/10$^6$ |
|---|---|---|---|---|---|---|
| **YOLOv5s** | 90.1 | 85.0 | 87.6 | 92.3 | 75.6 | 7.028 |
| **CBG-YOLOV5s** | 93.3 | 86.6 | 90.0 | 95.0 | 76.8 | 5.654 |
| **Comparison** | +3.2 | +1.6 | +2.4 | +2.7 | +1.2 | -1.374 |

evident that the proposed model outperformed other detection models in terms of accuracy in recognizing surface corrosion on metals. Considering all factors, our model exhibited superior detection results.

## 6.5. Ablation experiments

This paper introduced three enhancement methods, namely C3CBAM, BiFPN-CBAM, and C3Ghost. To verify the effectiveness of these modules for CBG-YOLOv5s in the detection of surface corrosion on metals, a series of ablation experiments were conducted. The results are presented in Table 8. In the table, "-" indicates that the module was not used, while "√" signifies that the module was incorporated.

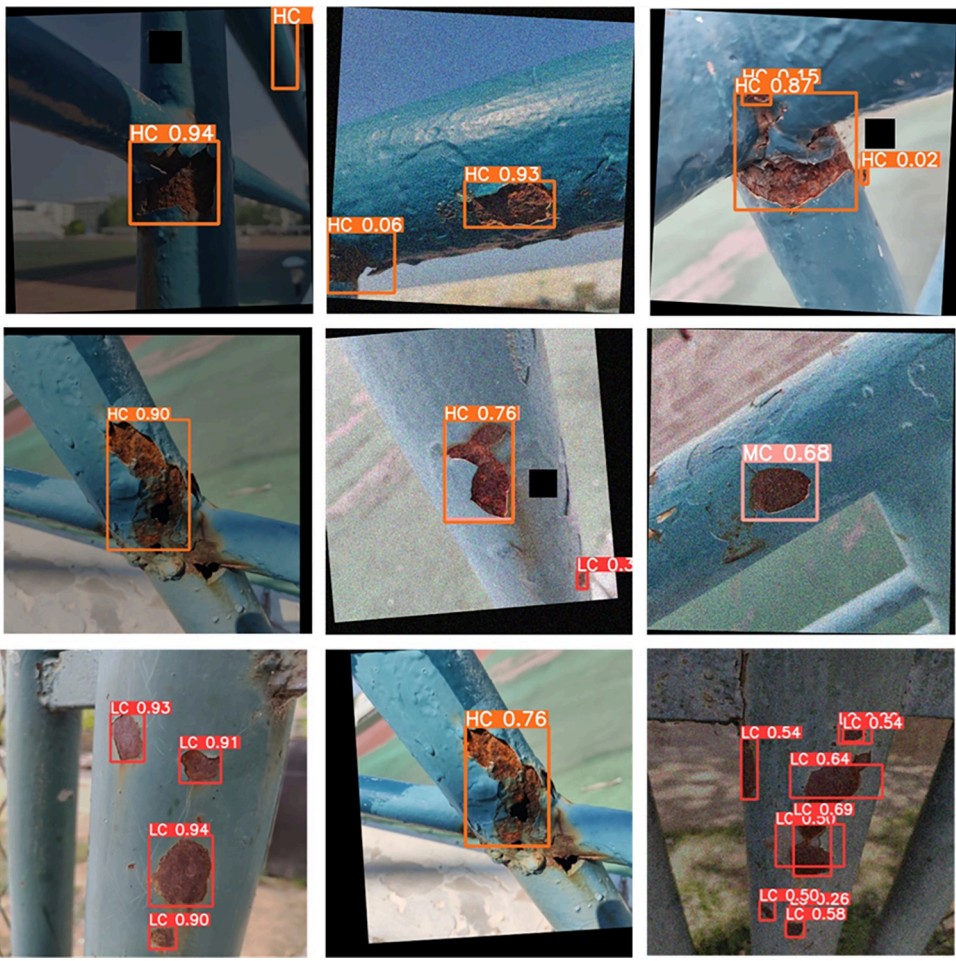

**Fig 11. YOLOv5s detection effect diagram.**

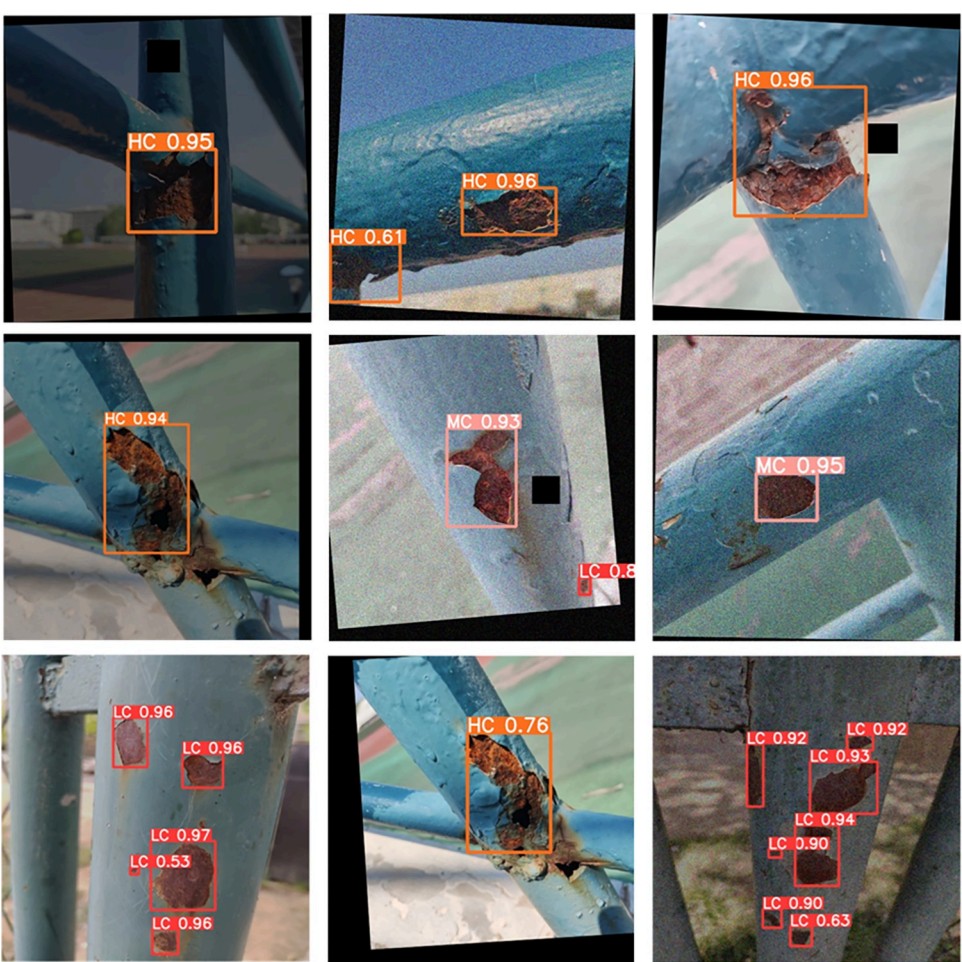

**Fig 12. CBG-YOLOv5s detection effect diagram.**

The experimental results demonstrated that the inclusion of the C3CBAM module led to a recall rate of 86.7%, validating its effectiveness in target localization. The introduction of the BiFPN-CBAM module resulted in an accuracy increase of 1.8% and an mAP increase of 0.8%, while also reducing the parameter volume to a certain extent. These findings suggest that adding a small target detection layer aids in identifying small targets within the dataset. The implementation of the C3Ghost module reduced the parameter volume by $0.736\times10^6$ and achieved an mAP@0.5 value of 95%. This opens up possibilities for deployment on embedded platforms.

**Table 7. Comparison with typical models.**

| Method | Backbone | Size | P/% | R/% | F1-score/% | mAP/% | FPS |
|---|---|---|---|---|---|---|---|
| CenterNet | ResNet50 | 512×512 | 91.6 | 86.1 | 88.8 | 87.0 | 69.2 |
| SSD | VGG16 | 300×300 | 87.2 | 60.4 | 71.4 | 69.5 | 86.1 |
| Faster R-CNN | ResNet50 | 600×600 | 64.5 | 80.1 | 71.5 | 79.7 | 13.0 |
| YOLOv3 | DarkNet53 | 416×416 | 79.4 | 63.4 | 70.5 | 67.9 | 64.8 |
| YOLOv4 | CSPDarkNet53 | 416×416 | 89.9 | 45.8 | 60.7 | 66.0 | 50.0 |
| YOLOv5s | CSPDarkNet | 640×640 | 90.1 | 85.0 | 87.5 | 92.3 | **169.5** |
| CBG-YOLOV5s | CSPDarkNet | 640×640 | **93.3** | **86.6** | **89.8** | **95.0** | 135.1 |

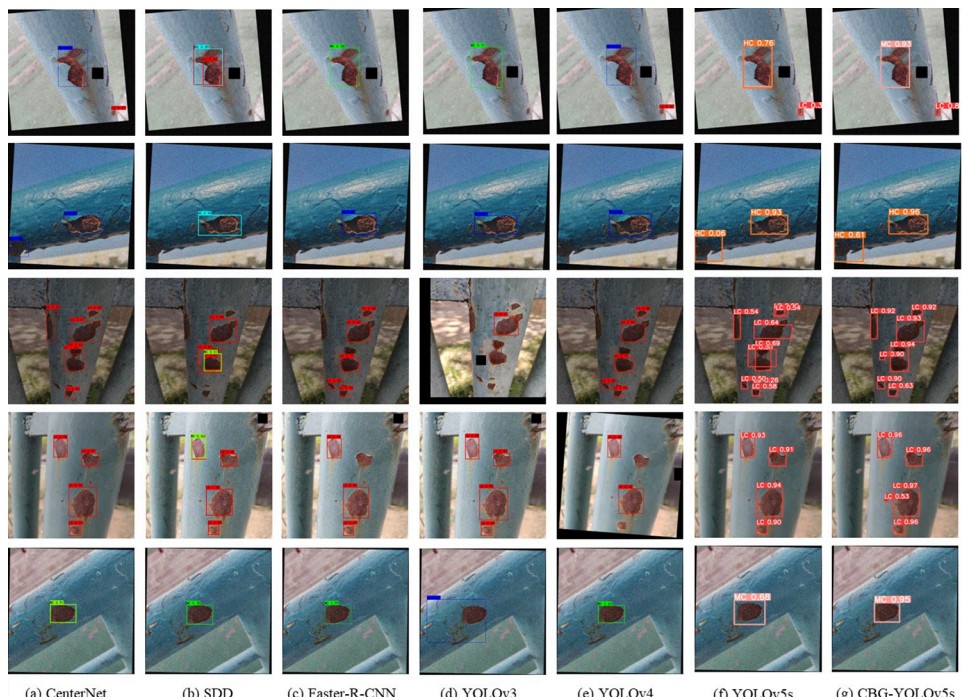

(a) CenterNet   (b) SDD   (c) Faster-R-CNN   (d) YOLOv3   (e) YOLOv4   (f) YOLOv5s   (g) CBG-YOLOv5s

**Fig 13. Recognition effect of each model.**

We utilised four metrics, namely precision, recall, mAP@.5 and mAP@.5:.95, to assess the behaviour of the CBG-YOLOv5s model. Fig 14 illustrates the variation curves of these evaluation metrics throughout the training procedure. As can be seen from the graph, the trends of these four evaluation metrics were largely similar. In the early stages of model training, all metrics increased rapidly, while in the later stages of training they tended to stabilise. After 200 training rounds, all metrics converged.

The experimental results substantiated that the enhancement strategies we proposed not only significantly improved the detection accuracy of the model for identifying surface corrosion on metals, but also effectively reduced the parameter volume. These achievements fully validated the effectiveness of the improvement methods we proposed.

## 6.6. Limitation and discussion

Although our research produced some results, there were still shortcomings. The main problem was that we collected a small number of corrosion materials in our data set, which was not representative. This is because different metal materials have different shapes, colours, depths and ranges of corrosion. We also needed to develop a complete system for detecting corrosion on metal surfaces so that it could be used in practical engineering.

**Table 8. Ablation experiments.**

| Method | C3CBAM | BiFPN-CBAM | C3Ghost | P/% | R/% | F1-score/% | mAP@0.5 | Params/10⁶ |
|---|---|---|---|---|---|---|---|---|
| YOLOv5s | - | - | - | 90.1 | 85 | 87.5 | 92.3 | 7.028 |
| YOLOv5s-1 | √ | - | - | 88.6 | **86.7** | 87.6 | 92.7 | 7.036 |
| YOLOv5s-2 | √ | √ | - | 90.4 | 86.3 | 88.3 | 93.6 | 6.390 |
| CBG-YOLOv5s | √ | √ | √ | **93.3** | 86.6 | **89.8** | **95** | **5.654** |

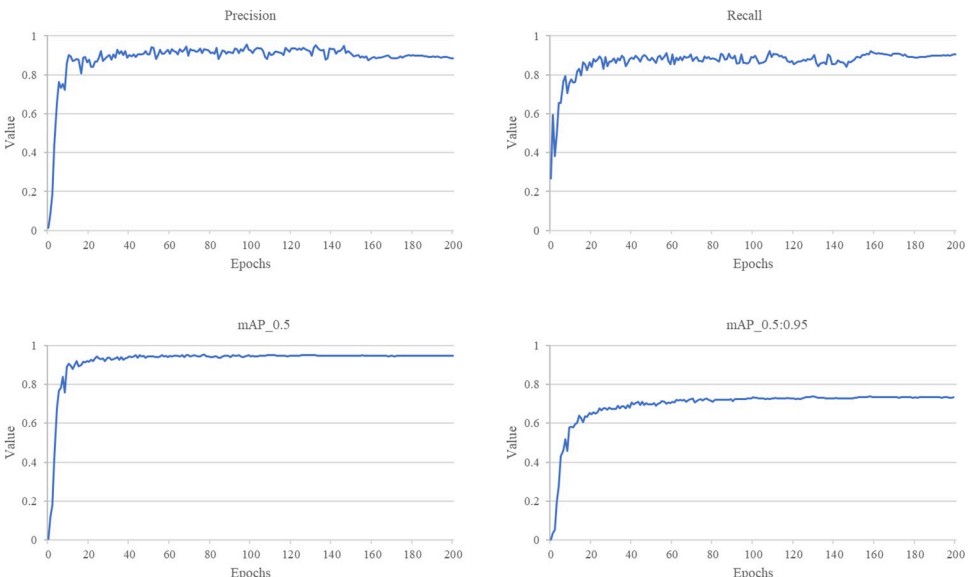

**Fig 14. Curve of changes in each evaluation indicator during the training process.**

To solve these problems, our next goal was to expand our data set, collect as many corrosion images of different metal materials as possible, and improve the generalisation ability of the CBG-YOLOv5s model so that it could adapt to more detection scenarios. Of course, this would also reduce our model detection speed, so we would further optimise the model structure and achieve model lightweighting on the basis of ensuring model detection accuracy. In addition, we would develop a complete detection system for this model and integrate it into handheld devices so that it could be used in real-life scenarios.

## 7. Conclusion

This paper proposed a novel model, CBG-YOLOv5s, for the detection of surface corrosion on metals. The model was designed to address challenges such as high background complexity in the corrosion area, small category differences, and dense targets. Firstly, we designed the C3CBAM module to more effectively extract features of surface corrosion on metals. Secondly, we introduced the BiFPN-CBAM module to enhance the model's detection capability for targets of different scales and to aid in the accurate identification of small targets. Lastly, we designed a lightweight C3Ghost module to compress the parameter volume, making the entire model more compact. Compared to other target detection models, this method has higher accuracy and lightweight advantages in the task of detecting surface corrosion on metals. Future research will focus on two directions: first, we will collect more extensive images of surface corrosion on metals to further improve the generalizing ability of the CBG-YOLOv5s model, making it applicable to a wider range of detection environments; second, while ensuring the accuracy of the model detection, we will further achieve model lightweighting for embedding in mobile handheld devices.

## Author Contributions

**Conceptualization:** Zhendong Cui.

**Data curation:** Mingjiao Fu.

**Formal analysis:** Zhitong Jia.

**Funding acquisition:** Zhitong Jia.

**Investigation:** Mingjiao Fu.

**Methodology:** Mingjiao Fu.

**Project administration:** Mingjiao Fu.

**Software:** Mingjiao Fu.

**Supervision:** Zhendong Cui.

**Validation:** Lingzhi Wu.

**Writing – original draft:** Mingjiao Fu.

**Writing – review & editing:** Zhendong Cui.

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
