## [Decision Letter · Decision Letter 0]

23 Nov 2023

PONE-D-23-36733Detection and recognition of metal Surface Corrosion based on CBG-YOLOv5sPLOS ONE

Dear Dr. Cui,

Thank you for submitting your manuscript to PLOS ONE. After careful consideration, we feel that it has merit but does not fully meet PLOS ONE’s publication criteria as it currently stands. Therefore, we invite you to submit a revised version of the manuscript that addresses the points raised during the review process.

We look forward to receiving your revised manuscript.

Kind regards,

Mohammed Abdelsamea, Ph.D

Academic Editor

PLOS ONE

Journal Requirements:

4. Please note that PLOS ONE has specific guidelines on code sharing for submissions in which author-generated code underpins the findings in the manuscript. In these cases, all author-generated code must be made available without restrictions upon publication of the work. Please review our guidelines at https://journals.plos.org/plosone/s/materials-and-software-sharing#loc-sharing-code and ensure that your code is shared in a way that follows best practice and facilitates reproducibility and reuse.

Additional Editor Comments:

There are several major issues that need to be addressed as raised by the reviewers, especially the unclear novelty of the proposed method and the lack of scientific discussion.

Reviewers' comments:

Reviewer's Responses to Questions

**Comments to the Author**

1. Is the manuscript technically sound, and do the data support the conclusions?

Reviewer #1: Yes

Reviewer #2: Partly

2. Has the statistical analysis been performed appropriately and rigorously? 

Reviewer #1: Yes

Reviewer #2: Yes

3. Have the authors made all data underlying the findings in their manuscript fully available?

Reviewer #1: Yes

Reviewer #2: No

4. Is the manuscript presented in an intelligible fashion and written in standard English?

Reviewer #1: Yes

Reviewer #2: Yes

5. Review Comments to the Author

Reviewer #1: This paper proposes a novel method for detecting and recognizing metal surface corrosion based on a modified YOLOv5s. The method integrates a convolutional block attention module (CBAM) into the feature extraction and fusion network and uses a ghost convolution to reduce the model complexity and improve the speed. The authors also construct a dataset of 6000 images of metal surface corrosion and evaluate the performance of the proposed method on this dataset. However, some minor changes are needed that address the following:

.      What are the main differences between the proposed attention techniques and existing those attention methods.

.      What are the differences in techniques between the proposed method and existing methods?

Reviewer #2: This paper proposed CBG-YOLOv5s, for the detection of surface corrosion on metals. I have the following concerns:

1. The introduction is written well. Literature review also provides sufficient information about previous studies.

2. However, it would be nice if authors can provide related work section and a table to deeply analyze the methods used in previous studies and the drawbacks and advantages of similar studies.

3. I suggest to re-write the major contributions of this work in the introduction section. Because all contributions are not novel. These methods are already existed and used in many works. What is novelty of this work in deep learning field?

4. Please re-draw Figures and cite them accordingly. These figures quality is not good. Check copyright issue.

5. Please add a "Limitation and Discussion" section to give a limitation of the proposed method and future research gaps in this field.

6. How does this method work with higher resolution images?

7. Authors should also check the article for typo errors and English grammar.

8. I recommend to review the following papers:

Improved Classification Approach for Fruits and Vegetables Freshness Based on Deep Learning

6. PLOS authors have the option to publish the peer review history of their article (what does this mean?). If published, this will include your full peer review and any attached files.

Reviewer #1: No

Reviewer #2: **Yes: **Mukhriddin Mukhiddinov

---

## [Author Response · Author response to Decision Letter 0]

28 Dec 2023

Response to Reviewers

Dear editor and reviewers, 

Thank you for the comments concerning our manuscript entitled “Detection and recognition of Metal Surface Corrosion based on CBG-YOLOv5s” (Manuscript Number: PONE-D-23-36733). Those comments were valuable and have been very helpful during the revision of our manuscript. We have read the comments carefully and made corrections. We have uploaded the revised manuscript. Changes in the revised manuscript are highlighted in yellow. The responses to the reviewers' comments are summarised below.

We would like to thank you for allowing us to resubmit a revised copy of the manuscript. We hope that the revised manuscript will be accepted and published in the PLOS ONE.

Sincerely,

Zhendong Cui

December 25, 2023

Response to reviewer 1

1. What are the main differences between the proposed attention techniques and existing those attention methods. 

Response:

Thank you very much for your comment. 

The "C3CBAM Attention Mechanism" is an attention mechanism designed specifically for object detection tasks. It embeds the CBAM module into the C3 module, endowing each C3 module with the functionality of the attention mechanism. This enhances the model's attention capability during the process of feature extraction and fusion. The CBAM module is a convolutional attention mechanism module that combines channel attention and spatial attention. It has the ability to independently weight the channel and spatial dimensions of features, thereby augmenting the representational capability of the features.

In contrast to alternative attention methods, the "C3CBAM Attention Mechanism" exhibits a more comprehensive and flexible attention mechanism. It can calibrate and optimize the channel and spatial dimensions of features simultaneously, thereby enhancing the performance and effectiveness of the CBG-YOLOv5s model. Furthermore, the C3CBAM attention mechanism also improves the detection accuracy and speed of the CBG-YOLOv5s model. Compared to other attention mechanisms, it is more suitable for object detection tasks because it can better capture the location and size information of the target, while also reducing the redundant information in the feature map.

2. What are the differences in techniques between the proposed method and existing methods. 

Response:

Thank you very much for your comment. 

Compared to the existing methods, the method of this study has the following technical differences:

(1). Unlike most existing methods that use a single or no attention mechanism, this study’s method employs the C3CBAM module, enabling better utilization of feature map information and improved target detection performance. 

(2). This study’s method extends the scale of the YOLOv5s model, unlike most existing methods that use the original YOLOv5s model or models based on YOLOv4 or YOLOv3. As a result, it can better adapt to detecting targets at different scales, particularly small targets, offering clear advantages.

(3). This study’s method employs Ghost convolution in place of ordinary convolution, unlike most existing methods that use ordinary or other types of convolution. This approach reduces model complexity and computational load, improves operational speed and efficiency, and ensures feature map quality.

Response to reviewer 2

1. The introduction is written well. Literature review also provides sufficient information about previous studies. 

Response:

Thank you very much for your comment. 

Thank you very much for your affirmation of my work, it makes me very happy.

2. However, it would be nice if authors can provide related work section and a table to deeply analyze the methods used in previous studies and the drawbacks and advantages of similar studies. 

Response:

Thank you very much for your comment. 

This is a great suggestion. In the final section of the related work, we have added the methods used in previous studies and the advantages and disadvantages of similar studies. Please refer to the attachment for details of the form.

3. I suggest to re-write the major contributions of this work in the introduction section. Because all contributions are not novel. These methods are already existed and used in many works. What is novelty of this work in deep learning field. 

Response:

Thank you very much for your comment. 

We agree with this suggestion and have rewritten the main contributions of this work in the introduction section as follows:

(1). We have developed a corrosion detection model for metal surfaces, called CBG-YOLOv5s. This model can classify metal surfaces into three corrosion levels based on their texture, colour, and depth of the corrosion. It can assist technicians in accurately and promptly identifying the corrosion level of metals.

(2). We collected 600 original images of corroded metal surfaces and performed data augmentation, constructing a dataset of metal surface corrosion images containing 6000 images.

(3). In order to improve the detection accuracy of the model, we introduced the C3CBAM module and C3Ghost module, expanded the scale of the YOLOv5s model, and added a small target detection layer. Compared with several other commonly used object detection models, our method had achieved superior performance in terms of detection accuracy and speed.

4. Please re-draw Figures and cite them accordingly. These figures quality is not good. Check copyright issue. 

Response:

Thank you very much for your comment. 

We agree with your suggestion. Based on your suggestions, we have redrawn the diagrams and referenced them accordingly, and hope that they meet your expectations. Please refer to the attachment for details of the redrawn figures.

5. Please add a "Limitation and Discussion" section to give a limitation of the proposed method and future research gaps in this field. 

Response:

Thank you very much for your comment. 

We think it's a good proposal. We added a “Limitation and Discussion” section in the revised manuscript, with the following new content:

6.6 Limitation and Discussion

Although our research produced some results, there were still shortcomings. The main problem was that we collected a small number of corrosion materials in our data set, which was not representative. This is because different metal materials have different shapes, colours, depths and ranges of corrosion. We also needed to develop a complete system for detecting corrosion on metal surfaces in the future works so that it could be used in practical engineering. 

To solve these problems, our next goal was to expand our data set, collect as many corrosion images of different metal materials as possible, and improve the generalisation ability of the CBG-YOLOv5s model so that it could adapt to more detection scenarios. Of course, this would also reduce our model detection speed, so we would further optimise the model structure and achieve model lightweighting on the basis of ensuring model detection accuracy. In addition, we would develop a complete detection system for this model and integrate it into handheld devices so that it could be used in real-life scenarios.

6. How does this method work with higher resolution images. 

Response:

Thank you very much for your comment. 

When dealing with high-resolution images, the CBG-YOLOv5s normalizes the resolution of the input image to a uniform image resolution of 640×640 before feeding it to the network, thus enabling it to handle images of different resolutions. Of course, this method may result in the loss of some detail information when dealing with high-resolution images. Therefore, another method for handling higher resolution images is provided in the YOLOv5 version. We can use the YOLOv5l6.pt pre-training weights provided to us by the model to process high-resolution images, which employs a technique called Padded Resize, which adjusts the resolution of the input image to a multiple of 32, and therefore allows for the processing of higher resolution images. However, it should be noted that when dealing with higher resolution images, the model requires larger computational and memory requirements. In practical applications, these parameters may need to be adjusted according to specific application scenarios and requirements.

7. Authors should also check the article for typo errors and English grammar. 

Response:

Thank you very much for your comment. 

We apologize for the poor writing in our article. We have meticulously checked the article for typos and English grammar, and have made corrections in the erroneous areas. We hope that the revised article will be recognised.

8. I recommend to review the following papers: Improved Classification Approach for Fruits and Vegetables Freshness Based on Deep Learning.

Response:

Thank you very much for your comment. 

We think it's a good proposal. We took your advice and read this paper carefully, gaining a lot. We learned from the strengths of this paper and optimized our own. Meanwhile, the reference [23] in the revised manuscript cited the article “Improved Classification Approach for Fruits and Vegetables Freshness Based on Deep Learning”.

---

## [Editor Report · Decision Letter 1]

28 Feb 2024

Detection and recognition of metal Surface Corrosion based on CBG-YOLOv5s

PONE-D-23-36733R1

Dear Dr. Cui,

We’re pleased to inform you that your manuscript has been judged scientifically suitable for publication and will be formally accepted for publication once it meets all outstanding technical requirements.

Kind regards,

Mohammed Abdelsamea, Ph.D

Academic Editor

PLOS ONE
---

## [Editor Report · Acceptance letter]

26 Mar 2024

PONE-D-23-36733R1 

PLOS ONE

Dear Dr. Cui, 

I'm pleased to inform you that your manuscript has been deemed suitable for publication in PLOS ONE. Congratulations! Your manuscript is now being handed over to our production team.

Kind regards, 

on behalf of

Dr. Mohammed Abdelsamea 

Academic Editor

PLOS ONE